# Factors Associated with the Development of Coagulopathy after Open Traumatic Brain Injury

**DOI:** 10.3390/jcm11010185

**Published:** 2021-12-30

**Authors:** Yuhui Chen, Jun Tian, Bin Chi, Shangming Zhang, Liangfeng Wei, Shousen Wang

**Affiliations:** Department of Neurosurgery, 900th Hospital, Fuzong Clinical Medical College of Fujian Medical University, Fuzhou 350025, China; cyh0608@163.com (Y.C.); Junt2001@163.com (J.T.); cbdahuaidan@163.com (B.C.); zhangshangming2019@163.com (S.Z.); wlf74@163.com (L.W.)

**Keywords:** open traumatic brain injury, coagulopathy, prognosis, PLR, TXA

## Abstract

Background: The incidence of coagulopathy after open traumatic brain injury (TBI) is high. Coagulopathy can aggravate intracranial hemorrhage and further increase morbidity and mortality. The purpose of this study was to determine the clinical characteristics of coagulopathy after open TBI and its relationship with the prognosis. Methods: This study retrospectively evaluated patients with isolated open TBI from December 2018 to December 2020. Coagulopathy was defined as international normalized ratio (INR) > 1.2, activated thromboplastin time (APTT) > 35 s, or platelet count <100,000/μL. We compared the relationship between the clinical, radiological, and laboratory parameters of patients with and without coagulopathy, and the outcome at discharge. Logistic regression analysis was used to evaluate the risk factors associated with coagulopathy. We then compared the effects of treatment with and without TXA in open TBI patients with coagulopathy. Results: A total of 132 patients were included in the study; 46 patients developed coagulopathy. Patients with coagulopathy had significantly lower platelet levels (170.5 × 10^9^/L vs. 216.5 × 10^9^/L, *p* < 0.001), and significantly higher INR (1.14 vs. 1.02, *p* < 0.001) and APTT (30.5 s vs. 24.5 s, *p* < 0.001) compared to those with no coagulopathy. A Low Glasgow Coma Scale (GCS) score, high neutrophil/lymphocyte ratio (NLR), low platelet/lymphocyte ratio (PLR), and hyperglycemia at admission were significantly associated with the occurrence of coagulopathy. Conclusions: Coagulopathy often occurs after open TBI. Patients with a low GCS score, high NLR, low PLR, and hyperglycemia at admission are at greater risk of coagulopathy, and therefore of poor prognosis. The efficacy of TXA in open TBI patients with coagulopathy is unclear. In addition, these findings demonstrate that PLR may be a novel indicator for predicting coagulopathy.

## 1. Introduction

Traumatic brain injury (TBI) is one of the most common causes of death and disability worldwide, and can be divided into primary brain injury acting directly on the skull and secondary brain injury caused by initial trauma [1]. Traumatic intracranial hemorrhage (epidural, subarachnoid, arachnoid, or parenchymal) occurs in approximately 50% of patients; further, TBI often results in coagulopathy, which can aggravate intracranial hemorrhage and further increase mortality and morbidity [2]. Previous studies have confirmed that coagulopathy occurs more frequently in open TBI than in closed TBI [2,3], which may be associated with the more severe brain injury in open TBI [4,5].

Coagulopathy can be divided into a hypocoagulable state characterized by prolonged bleeding and bleeding progression, and a hypercoagulable state characterized by an increased risk of thrombosis. Although both of these states can co-exist after TBI, a hypocoagulable state tends to be more common [6]; nevertheless, the clinical significance, pathophysiological mechanisms, and temporal relationship of these two phenotypes are unknown [6,7]. Several theories have been proposed to explain the underlying mechanism. One theory is based on the fact that brain tissue contains the highest prothrombin content in the human body, and TBI leads to the release of large amounts of the tissue factor from the microvessels of the damaged brain tissue into the bloodstream, resulting in activation of the coagulation cascade through the external coagulation pathway and insufficient consumption of coagulation factors [7,8,9]. In addition, the interaction between shock and hypoperfusion leads to the activation of protein C, which promotes apoptosis by inactivating proenzyme activator inhibitor-1 [7]. A recent prospective study [10] demonstrated that isolated TBI is consistent with coagulation changes in non-TBI patients and is characterized by disseminated intravascular coagulation with hyperactivity, that is, increased plasmin production and decreased antiplasmin levels. However, these hypotheses lack strong clinical evidence, and the role of inflammatory parameters remains unclear. Despite its high incidence and severe adverse outcomes, there are limited data regarding the mechanism of development of coagulopathy after open TBI. Additional data is essential, as a clear understanding of risk factors is crucial to assess the severity of injury and identify patients with good prognosis.

In this retrospective study, we aimed to determine the clinical features and risk factors of coagulopathy after open TBI. In addition, we analyzed the relationship between coagulopathy and adverse functional outcomes.

## 2. Materials and Methods

All subjects gave their informed consent for inclusion before they participated in the study. The study was conducted in accordance with the Declaration of Helsinki, and the protocol was approved by the Ethics Committee of the 900th hospital.

## 3. Patient Population

This was an observational, retrospective cohort study. Patients admitted to the neurosurgery department of the 900th Hospital between December 2018 and December 2020 were selected. All patients underwent surgery and were performed by the same group of surgeons. The inclusion criteria were as follows: (1) diagnosis of open TBI, defined as an injury in which the injured object results in the opening of the scalp, skull, dura, and brain tissue to the outside world, (2) isolated TBI, without concomitant injuries, and (3) age ≥16 years. Exclusion criteria were: (1) death on admission, (2) absence of imaging or laboratory tests within 3 h of admission, (3) absence of evidence of dural damage on imaging, and (4) prior use of anticoagulants or antiplatelet drugs or history of coagulopathy.

## 4. Data Collection

The demographic and clinical data collected the included age, sex, mechanism of injury, Glasgow Coma Scale (GCS) score on admission, results of computed tomography (CT) scan of the head on admission, laboratory tests (hemoglobin (Hb), neutrophil/lymphocyte ratio (NLR), platelet/lymphocyte ratio (PLR), platelet count (PLT), international normalized ratio (INR), and activated thromboplastin time (APTT)), admission glucose level, preoperative tranexamic acid (TXA) treatment, the time from open TBI to the surgery, decompressive craniectomy (DC), operative blood loss, duration of surgery, number of blood transfusions, blood transfusion volume, and length of ICU stay (ICU LOS), and number of deaths.

Outcome was measured using the Glasgow Outcome Scale (GOS) at discharge. Under this rating system, a GOS score of 1 indicates death, 2 indicates a persistent vegetative state, 3 indicates severe disability (conscious but disabled), 4 indicates moderate disability (disabled but independent), and 5 indicates excellent recovery with a return to the baseline functional status.

## 5. Definition of Coagulopathy

In the absence of a generally accepted definition of coagulopathy after TBI, coagulopathy was defined as any of the following conditions based on previous studies [6,7,8,9], and the definitions and reference values proposed by local institutions and laboratories: INR > 1.2, APTT > 35 s, or PLT < 100,000/μL. All CT and laboratory data were obtained within 3 h of admission.

## 6. Statistical Analysis

Continuous variables were expressed as means with standard deviation (SD) or medians with interquartile ranges (IQR = Q3–Q1), and categorical variables were expressed as percentages. According to the above criteria, all patients were divided into two groups: coagulopathy and no coagulopathy. Comparisons between groups were made using the Mann–Whitney U test or two-sample t-test. Variables with a *p* value of <0.05, as determined by univariate analysis, were included in the binary multivariate logistic regression analysis to derive the potential factors independently associated with the development of coagulopathy or unfavorable functional outcome. All data were statistically analyzed using SPSS 20.0 (IBM, New York, NY, USA), and *p* < 0.05 was considered statistically significant.

## 7. Results

### 7.1. Subject Recruitment

A total of 202 open patients with TBI were reviewed in this study, of which 70 were excluded due to the following reasons: age < 16 years or unknown (*n* = 13), polytrauma (*n* = 29), death during initial resuscitation (*n* = 6), missing initial imaging and laboratory tests (*n* = 7), radiographic evidence of lack of dural opening (*n* = 5), and prior use of anticoagulants or antiplatelet drugs (*n* = 10). Finally, 132 patients were included in the analysis, of which 46 (34.8%) patients had coagulopathy, and 86 (65.2%) did not (Figure 1).

### 7.2. Basic Characteristics of the Population

The age (interquartile range [IQR]) of our cohort was 44.5 years (33.3–57 years), and 113 (85.6%) patients were male. Blow injury (43 patients; 32.6%) was the most frequent mechanism of injury; traffic accidents were also common (40 patients; 30.3%), followed by free-fall (23 patients; 17.4%) and fall (20 patients; 15.9%). The median GCS score (IQR) at admission was 10 (8–14). Fifty patients (37.9%) had skull vault fractures alone, 21 (15.9%) had skull base fractures alone, and 61 (46.2%) had both fractures. Venous sinus injury was found in 19 cases (14.4%), intracranial fragments in 11 cases (8.3%), intracranial hematoma in 49 cases (37.1%), epidural hematoma in 65 cases (49.2%), subdural hematoma in 50 cases (37.9%), subarachnoid hemorrhage in 73 cases (55.3%), and midline shift in 34 cases (25.8%) on head CT. The median (IQR) of serum laboratory tests on admission were as follows: Hb, 131.5 g/L (121–141.8 g/L); NLR, 15.2 (9.4–19.1); PLR, 160.2 (92.1–244.3); PLT, 203 × 10^9^/L (165–243 × 10^9^/L); INR, 1.05 (0.99–1.13); APTT, 25.3 s (23.1–29.8 s); and glucose, 8.5 mmol/L (6.5–10.6 mmol/L). Twenty-two (16.7%) patients received TXA after admission, the mean time to surgery after TBI was 7.5 h (±2.8 h), 37 (28.0%) patients underwent decompressive craniectomy, the mean operative blood loss was 402 mL (±408 mL), the mean operation time was 147 min (±86 min), 60 (45.5%) suffered blood transfusion, the mean blood transfusion volume was 546 mL (±804 mL), and the mean ICU length of stay was 4.0 d (±4.7 d). The median GOS (IQR) was 3 (3–5) and there were 21 deaths (15.9%) (Table 1).

Typical preoperative and postoperative CT images of an open TBI patient with coagulopathy are shown in Figure 2.

### 7.3. Comparison of Coagulopathy and No Coagulopathy Groups

In univariate analysis, age (IQR) was 51.5 years (34.3–69.0 years) in the coagulopathy group and 44.0 years (32.8–56.0 years) in the no coagulopathy group. The coagulopathy group had a higher proportion of males (40 patients, 87.0%) compared to the no coagulopathy group (73 patients, 84.9%). There was no significant difference in age or sex between the two groups. In addition, patients with low GCS were more likely to develop coagulopathy (*p* < 0.001), and patients with coagulopathy had more frequent intracranial hematoma (*p* < 0.001), subdural hematoma (*p* < 0.001), subarachnoid hemorrhage (*p* < 0.001), and midline shift (*p* = 0.004). No significant differences were observed between the groups in terms of the mechanism of injury (*p* = 0.404), injury location (*p* = 0.128), venous sinus injury (*p* = 0.449), intracranial fragments (*p* = 0.747), or the presence of epidural hematoma (*p* = 0.089). For the coagulopathy cohort, the median INR (IQR) was 1.14 (1.08–1.30), the median APTT (IQR) was 30.5 s (24.8–35.73 s), and the median PLT (IQR) was 170.5 × 10^9^/L (145.3–205.3 × 10^9^/L). The median INR (IQR) was 1.02 (0.97–1.08), APTT (IQR) was 24.5 s (23.0–26.8 s), and PLT (IQR) was 216.5 × 10^9^/L (182.5–249.0 × 10^9^/L) in the no coagulopathy cohort. There were significant differences in all three coagulation variables between the coagulopathy and no coagulopathy groups (all *p* < 0.001). In addition, patients with coagulopathy had significantly lower Hb (*p* = 0.006), higher NLR (*p* < 0.001), lower PLR (*p* < 0.001), and higher admission glucose (*p* < 0.001). The coagulopathy group appeared to be more frequently treated with TXA (*p* = 0.034), decompressive craniectomy (*p* < 0.001), and blood transfusion (*p* < 0.001). In addition, patients with coagulopathy required more time for surgery (*p* = 0.029), blood loss (*p* < 0.001), blood transfusion volume (*p* < 0.001), and ICU LOS (*p* = 0.003). The length of time from TBI to surgery did not differ significantly between the two groups (*p* = 0.114) (Table 1).

Multivariate logistic regression analysis showed that the admission GCS score (OR = 0.720, 95% CI = 0.536–0.967, *p* = 0.029), NLR (OR = 1.653, 95% CI = 1.272–2.149, *p* < 0.001), PLR (OR = 0.971, 95% CI = 0.957–0.986, *p* < 0.001) and hyperglycemia (OR = 1.404, 95% CI = 1.060–1.860, *p* = 0.018) were independent risk factors for the development of coagulopathy, while intracranial hematoma (OR = 0.828, 95% CI = 0.157–4.371, *p* = 0.824), subdural hematoma (OR = 3.175, 95% CI = 0.546–18.461, *p* = 0.198), subarachnoid hemorrhage (OR = 0.227, 95% CI = 0.023–2.283, *p* = 0.208), midline shift (OR = 0.528, 95% CI = 0.088–3.156, *p* = 0.484), and Hb (OR = 0.977, 95% CI = 0.941–1.015, *p* = 0.236) had no significant association with the development of coagulopathy (Table 2).

### 7.4. Comparison of Receiving and Not Receiving TXA in the Coagulopathy Group

We compared the results of the coagulopathy cohort treated with TXA with those of patients not treated with TXA (cases 12 vs. 34). The results showed that there was no significant difference in admission GCS scores between the two groups (*p* = 0.186). Although a higher proportion of patients treated with TXA were transfused (100% vs. 70.6%) and had a larger volume of blood transfused (1438 vs. 917 mL), no significant difference was observed (*p* = 0.086, 0.170, respectively). In addition, no significant differences were found in DC (*p* = 0.502), mean blood loss (*p* = 0.217), duration of surgery (*p* = 0.910), ICU LOS (*p* = 0.187), and prognosis (*p* = 0.639) (Table 3).

### 7.5. Outcomes Associated with Coagulopathy

In terms of outcomes, the median GOS score (IQR) for the coagulopathy cohort was 3 (1–4) versus 5 (4–5) in the no coagulopathy cohort (*p* < 0.001). There was a significant difference in mortality between the two cohorts (37% vs. 4.7%, *p* < 0.001) (Table 1). In addition, patients who developed coagulopathy had significantly lower prognostic scores at discharge than those who did not (*p* < 0.001) (Table 4). In the different GOS score groups, the lower the GOS score, the higher the proportion of patients with coagulopathy (Figure 3).

## 8. Discussion

This study included 132 patients and revealed the characteristics and outcomes of coagulopathy in patients with open TBI. According to the definition employed, approximately one-third of patients experienced coagulopathy. Patients with coagulopathy had lower GCS and GOS scores and higher mortality than those without coagulopathy, which suggests a correlation between coagulation parameters and open TBI severity and prognosis. Multivariate regression analysis revealed that coagulopathy was associated with GCS score, NLR, PLR, and hyperglycemia.

The GCS score reflects the severity of brain injury, and a lower GCS score is associated with the occurrence of coagulopathy and can be used as a predictor of mortality [2,6,7,11]. These results are consistent with our own observation of a significantly lower GCS score in patients with coagulopathy than in those with no coagulopathy, suggesting that patients suffering from severe TBI are more likely to experience coagulopathy. However, previous studies were limited by the fact that they did not exclude patients with injuries elsewhere in the body [11], or because they only included patients with gunshot wounds [2]. By comparison, all patients in this study had isolated open TBI, which provides more robust evidence that brain injury severity is a risk factor for the development of coagulopathy.

Inflammatory measures, such as NLR [12,13,14,15] and PLR [16,17], play an important role in predicting the clinical outcomes of neurotraumatic diseases. Numerous studies have reported that a high NLR on admission is associated with poor prognosis in patients with neurological injury [14,15,18]. However, to the best of our knowledge, our study is the first to use NLR as a predictor of coagulopathy after open TBI. We believe that patients with coagulopathy have more severe brain injuries and more severe acute inflammatory reactions than patients without coagulopathy. The NLR is a composite of neutrophil and lymphocyte counts. TBI results in a more significant elevation of neutrophils, an indicator of the severity of the acute inflammatory response, than other leukocytes. By contrast, lymphocytes are elevated in response to chronic inflammation or viral infection, and therefore, lymphocyte counts may not be significantly increased after acute TBI [12]. Thus, the elevation in neutrophil count likely contributed to the elevated NLR of patients with coagulopathy, which was significantly higher than that of patients without coagulopathy (*p* < 0.001). The mechanism underlying the elevation of neutrophils in TBI may be related to damage-associated molecular patterns released after central nervous system injury, which may initially trigger an inflammatory response [19], but how it affects coagulation function remains unclear.

Previous studies have also demonstrated the value of the PLR as a prognostic indicator of the inflammatory response in conditions such as pulmonary embolism [20], stroke [21], and cancer [22]. The findings of these studies mostly indicate an association between PLR and the state of blood hypercoagulability. Although TBI is characterized by acute inflammation, there is almost no evidence that PLR influences the severity and prognosis of TBI. Xie et al. [16] found that the increase in PLR after traumatic spinal cord injury indicates a severe inflammatory response, which can effectively help identify patients with poor prognosis. In a study of 183 patients with intracranial hemorrhage, Zhang et al. [23] found that high PLR on admission was associated with worse GCS scores. They concluded that PLR was superior to PLT or lymphocyte counts alone in predicting neurological outcome, and is a more accurate indicator of a high level of inflammation in patients with cerebral hemorrhage. On the contrary, a recent prospective study by Idowu et al. [24] showed that a low PLR on admission indicated coagulopathy, PLT dysfunction, or thrombocytopenia, and that PLR was decreased in all patients who died, and could be used as an indicator of poor prognosis in chronic subdural hematoma. This is consistent with the results of the present study. While elevated PLR suggests a high level of inflammatory response in some studies, this may not be the case in patients with hemorrhagic brain injury. In our study, we observed a lower PLR in patients with coagulopathy after TBI, which may be due to the massive release of tissue factors after severe open TBI, resulting in activation of the coagulation cascade, PLT consumption, and a subsequent reduction in PLR; however, the incidence and underlying mechanism remain unclear. In addition, a recently published review by Bradbury et al. [25] suggests that platelet dysfunction is the most important cause of coagulopathy in TBI patients, including dysfunction of platelet adhesion, activation, and aggregation, but whether platelet transfusion can improve patient outcomes remains controversial. To the best of our knowledge, our study is the first to report the PLR as a predictor of coagulopathy after TBI. Future studies are warranted to determine the prognostic value of PLR, a new inflammatory indicator in TBI.

Our study found that hyperglycemia on admission was an independent risk factor for coagulopathy in open TBI (OR = 1.345, 95% CI = 1.052–1.720, *p* < 0.018). Yuan et al. [26] investigated the role of blood glucose levels on admission after TBI and reported that elevated blood glucose levels were associated with increased mortality and poor prognosis, and that admission blood glucose levels were significantly elevated in patients with coagulopathy. Alexiou et al. [9] found that blood glucose levels were associated with TBI severity as assessed by the GCS score, and coagulopathy may occur in TBI patients with blood glucose levels ≥15.1 g/L on admission. Although hyperglycemia after TBI may indicate the severity of injury and poor prognosis, there is no consensus on whether intensive blood glucose control is beneficial in patients with TBI [27,28]. We propose that the stress response after severe TBI leads to increased levels of catecholamines and decreased insulin secretion, which may lead to hyperglycemia. Hyperglycemia can enhance the expression of inflammatory factors and play a key role in promoting coagulopathy. Consequently, it is necessary to conduct prospective clinical studies on the relationship between blood glucose levels and coagulopathy in patients with open TBI to determine whether blood glucose level control can reduce the incidence of coagulopathy and improve patient outcomes.

Early correction of coagulopathy is of great significance for better survival of TBI, and treatment of coagulopathy mainly includes transfusion of fresh frozen plasma (FFP), platelets, and use of TXA. Although animal experiments have demonstrated that infusion of FFP can down-regulate the expression of inflammatory pathways and reduce brain swelling [29], Anglin et al. [30] showed that TBI patients with moderate coagulopathy who underwent FFP were more likely to have a poor prognosis. Thrombocytopenia and platelet dysfunction are central to the development of coagulopathy in TBI patients [25], platelet transfusion has not been shown to improve mortality in patients with intracranial hemorrhage after TBI, but may instead increase transfusion-related risks in patients with mild TBI, including allergic reactions, acute lung injury, and infections [31]. In our study, patients presenting with coagulopathy appeared to be more frequently treated with TXA (*p* = 0.034), DC (*p* < 0.001), and transfusion (*p* < 0.001), as well as more transfused volume (*p* < 0.001). This may be due to the fact that the patients in this study had high intraoperative blood loss due to severe coagulopathy and required massive blood transfusion to maintain blood pressure stability. However, it is unclear whether patients with significant blood loss and established severe coagulopathy require transfusion, including FFP and platelets [30]. TXA reduces bleeding by resisting fibrinolysis. The CRASH-3 trial [32] showed that TXA reduced the risk of craniocerebral injury-related death in patients with mild to moderate TBI compared with severe TBI, and that early treatment was better. The results of another randomized controlled trial [33] in patients with moderate to severe TBI showed that TXA given within 2 h did not play a favorable role in mortality, prognostic outcomes, and bleeding progression. Due to the lack of TXA assessment for TBI craniotomy patients, the prospective trial by Wu et al. [34] is trying to clarify the efficacy and safety of TXA for TBI patients undergoing craniotomy and determine its impact on the prognosis. Here, our study analyzed the intraoperative conditions and outcomes of patients treated with TXA in the coagulopathy group, and similar to the results of previous studies, we did not find that preoperative use of TXA improves outcomes as well as reduces blood transfusion in patients with severe open TBI with coagulopathy. Therefore, further studies are needed in the future to confirm the mortality and prognostic impact of blood transfusion and TXA in severe TBI patients with severe coagulopathy. However, TXA is still a potential treatment for patients with mild to moderate TBI.

Different degrees of coagulopathy after TBI have been repeatedly associated with lower GOS scores, but the morbidity and mortality rates reported by different studies vary considerably [6]. This may be due to the definitions of coagulopathy used in different studies, including different combinations of INR, platelet (PLT), prothrombin time (PT), prothrombin ratio/rapid (PTR), APTT, fibrinogen, and d-dimer positivity [6]. Van Gent et al. [7] defined coagulopathy as INR > 1.2 and/or PLT < 150 × 10^9^/L and/or APTT > 34.5 s. They found that the incidence of post-traumatic coagulopathy was 27%. Joseph et al. [35] analyzed the clinical characteristics of 591 TBI patients and showed that the incidence of coagulopathy patients was 13.3%, which was lower than the study by van Gent et al. [7], possibly due to the stricter definition of coagulopathy by this study, including INR ≥ 1.5 or PLT ≤ 100 × 10^9^/L or APTT ≥ 35 s. In a recent review by Epstein et al. [5], 22 retrospective and prospective studies were analyzed to examine the incidence and clinical outcome of coagulopathy in an isolated TBI population. The authors found that the incidence of coagulopathy was 35.2%, which is similar to the results of this study (34.8%), but the mortality rate varied widely, from approximately 17–86%, while the mortality rate in this study was 37%. These findings suggest that a standardized definition of coagulopathy needs to be established to reduce the heterogeneity between studies. At the same time, this study found that patients with isolated open TBI and coagulopathy had a longer duration of surgery (*p* = 0.029), longer ICU stay (*p* = 0.003), worse prognosis (GOS 3 vs. 5), and higher mortality (37% vs. 4.7%) than those without coagulopathy. Therefore, monitoring coagulation function in patients with open TBI can indicate the degree of brain tissue injury and help evaluate patient prognosis.

Our results clearly indicate that the severity of brain injury, inflammatory markers, and hyperglycemia play an important role in the development of coagulopathy in patients with open TBI. However, we did not find an effect of TXA on the outcome of open TBI patients with coagulopathy. In addition, our study subjects were all patients with isolated open TBI, which ruled out coagulopathy caused by injuries to other parts of the body. Further, this is the first study to report the PLR as a predictor of coagulopathy after open TBI.

Nevertheless, this study has some limitations worth noting. First, this was a single-center retrospective study with a small sample size. In the future, larger multicenter studies are required to validate these results and predict the risk of coagulopathy after open TBI. Second, the definition of coagulopathy lacks a unified standard. In this study, the definition was based on local experimental institutions and previous research. In future studies, a unified definition of coagulopathy may be needed. Finally, we only evaluated the GOS score at discharge to measure outcome; therefore, the effect of coagulopathy on the long-term prognosis of patients remains unknown. Further research is needed to clarify the role of coagulation disorders in open TBI.

In summary, patients who develop coagulopathy after open TBI have a worse prognosis and higher mortality rates than those who do not. Patients with a low GCS score, high NLR, low PLR, and hyperglycemia on admission were more likely to develop coagulopathy. The role of TXA in the prognosis of open TBI patients with coagulopathy remains unclear. In addition, PLR may be a new clinical indicator for predicting risk factors related to bleeding after TBI.

## Figures and Tables

**Figure 1 jcm-11-00185-f001:**
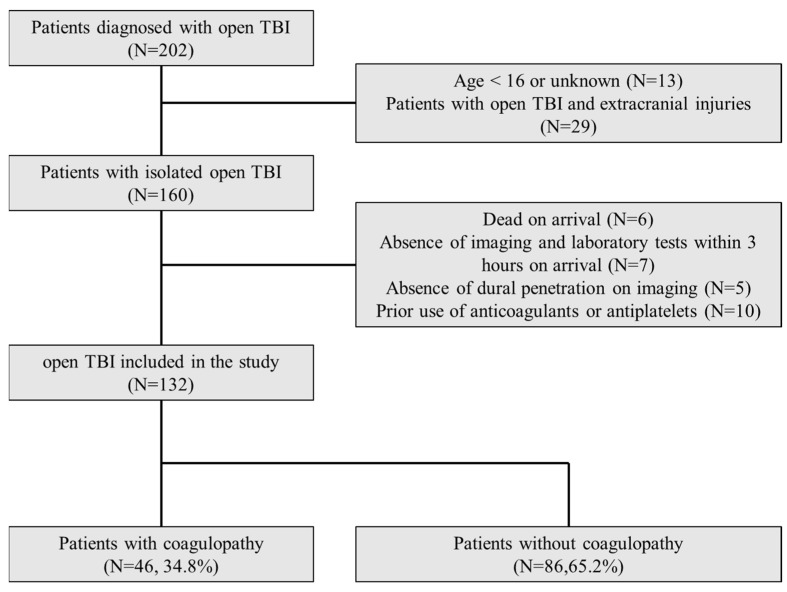
Flow chart of patient selection. TBI: traumatic brain injury.

**Figure 2 jcm-11-00185-f002:**
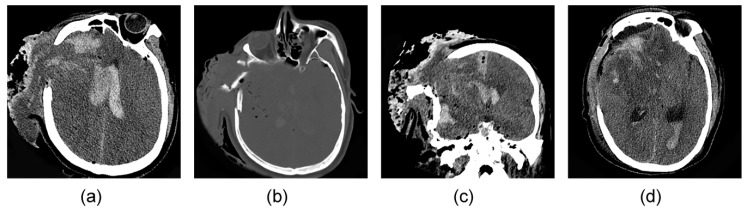
CT images of an open TBI patient with coagulopathy showing brain tissue extravasation, arachnoid hemorrhage and intracranial hematoma in the transverse section (**a**), comminuted fracture of right temporal bone and pneumocephalus in the bone window (**b**), subdural hematoma in coronal section (**c**), and re-examination 1 d after decompressive craniectomy showing hematoma and ventricular hematocele (**d**).

**Figure 3 jcm-11-00185-f003:**
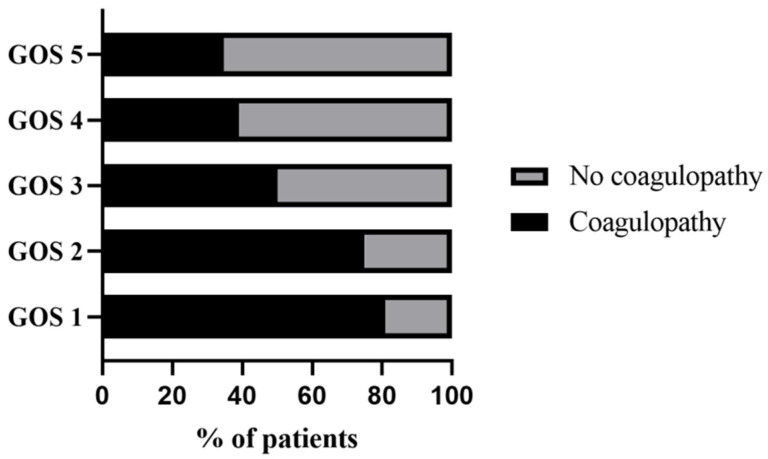
Comparison of the proportion of patients with coagulopathy in different prognostic scores.

**Table 1 jcm-11-00185-t001:** Comparison of baseline demographics and clinical characteristics of patients with and without coagulopathy after open TBI.

Variables	All Patients (*n* = 132)	Coagulopathy (*n* = 46)	No Coagulopathy (*n* = 86)	*p* Value
Median age (IQR) yrs.	44.5 (33.3–57)	51.5 (34.3–69.0)	44.0 (32.8–56.0)	0.076
Male (%)	113 (85.6)	40 (87.0)	73 (84.9)	0.801
Mechanism of injury (%)				0.404
Traffic accident	40 (30.3)	16 (34.8)	24 (27.9)	
Blow injury	43 (32.6)	11 (23.9)	32 (37.2)	
Fall	20 (15.2)	6 (13.0)	14 (16.2)	
Free fall	23 (17.4)	11 (23.9)	12 (14.0)	
Others	6 (4.5)	2 (4.4)	4 (4.7)	
Median GCS (IQR)	10 (8–14)	5.5 (4–9)	13 (10–15)	<0.001 *
Injury location (%)				0.128
Vault-SF	50 (37.9)	12 (26.1)	38 (44.2)	
Base-SF	21 (15.9)	9 (19.6)	12 (13.9)	
Both-SF	61 (46.2)	25 (54.3)	36 (41.9)	
Venous sinus injury (%)	19 (14.4)	5(10.9)	14 (16.3)	0.449
Intracranial fragments (%)	11 (8.3)	3(6.5)	8 (9.3)	0.747
Intracranial hematoma (%)	56 (42.4)	30 (65.2)	26 (30.2)	<0.001 *
EDH (%)	65 (49.2)	18 (39.1)	47 (54.7)	0.089
SDH (%)	50 (37.9)	27 (58.7)	23 (26.7)	<0.001 *
SAH (%)	73 (55.3)	39 (84.8)	34 (39.5)	<0.001 *
Midline shift (%)	34 (25.8)	19 (41.3)	15 (17.4)	0.004 *
Median Hb (IQR)	131.5 (121–141.8)	124.0 (118.8–136.0)	136.5 (123.5–144.0)	0.006 *
Median NLR (IQR)	15.2 (9.4–19.1)	18.7 (15.8–21.6)	11.9 (7.9–16.6)	<0.001 *
Median PLR (IQR)	160.2 (92.1–244.3)	106.2 (65.0–161.0)	193.0 (138.2–276.9)	<0.001 *
Median PLT (IQR)	203 (165–243)	170.5 (145.3–205.3)	216.5 (182.5–249.0)	<0.001 *
Median INR (IQR)	1.05 (0.99–1.13)	1.14 (1.08–1.30)	1.02 (0.97–1.08)	<0.001 *
Median APTT (IQR)	25.3 (23.1–29.8)	30.5 (24.8–35.73)	24.5 (23.0–26.8)	<0.001 *
Admission Glucose(IQR)	8.5 (6.5–10.6)	11.3 (9.4–14.6)	7.1 (6.1–8.8)	<0.001 *
Receive TXA (%)	22 (16.7)	12 (26.1)	10 (11.6)	0.034 *
Mean time from open TBI to the surgery (SD)	7.5 (2.8)	7.1 (2.8)	7.7 (2.8)	0.114
DC (%)	37 (28.0)	23 (50.0)	14 (16.3)	<0.001 *
Mean operative blood loss (SD)	402 (408)	649 (499)	270 (272)	<0.001 *
Mean duration of surgery (SD)	147 (86)	167 (98)	137 (78)	0.029 *
Blood transfusion (%)	60 (45.5)	36 (78.3)	24 (27.9)	<0.001 *
Mean blood transfusion volume (SD)	546 (804)	1053 (1016)	275 (488)	<0.001 *
Mean ICU LOS (SD)	4.0 (4.7)	5.9 (6.1)	3.0 (3.5)	0.003 *
Median GOS(IQR)	3 (3–5)	3 (1–5)	5 (4–5)	<0.001 *
Mortality (%)	21 (15.9)	17 (37.0)	4 (4.7)	<0.001 *

Abbreviations: GCS, Glasgow Coma Scale; SF, skull fracture; EDH, epidural hematoma; SDH, subdural hematoma; SAH, subarachnoid hemorrhage; Hb, hemoglobin; NLR, neutrophil/lymphocyte ratio; PLR, platelet/lymphocyte ratio; PLT, platelets; INR, international normalized ratio; APTT, activated partial thromboplastin time; TXA, tranexamic acid; DC, decompressive craniectomy; ICU, Intensive Care Unit; LOS, Length of stay; GOS, Glasgow outcome scale; IQR, interquartile range; SD, standard deviation. * These values are significant for statistical analysis.

**Table 2 jcm-11-00185-t002:** Multivariate logistic regression analysis for patients with coagulopathy.

Variables	Standard Error	Odds Ratio	95% CI	*p* Value
Median GCS (IQR)	0.150	0.720	0.536–0.967	0.029 *
Intracranial hematoma (%)	0.849	0.828	0.157–4.371	0.824
SDH (%)	0.898	3.175	0.546–18.461	0.198
SAH (%)	1.178	0.227	0.023–2.283	0.208
Midline shift (%)	0.912	0.528	0.088–3.156	0.484
Median Hb (IQR)	0.020	0.977	0.941–1.015	0.236
Median NLR (IQR)	0.134	1.653	1.272–2.149	<0.001 *
Median PLR (IQR)	0.008	0.971	0.957–0.986	<0.001 *
Median PLT (IQR)	NA ^a^	NA ^a^	NA ^a^	NA ^a^
Median INR (IQR)	NA ^a^	NA ^a^	NA ^a^	NA ^a^
Median APTT (IQR)	NA ^a^	NA ^a^	NA ^a^	NA ^a^
Admission Glucose (IQR)	0.143	1.404	1.060–1.860	0.018 *

Abbreviations: GCS, Glasgow Coma Scale; SDH, subdural hematoma; SAH, subarachnoid hemorrhage; Hb, hemoglobin; NLR, neutrophil: lymphocyte ratio; PLR, platelets: lymphocyte ratio; PLT, platelets; INR, international normalized ratio; APTT, activated partial thromboplastin time; IQR, interquartile range. NA ^a^ = not analyzed due to medication itself being a defining factor for coagulopathy. * These values are significant for statistical analysis.

**Table 3 jcm-11-00185-t003:** Comparison of surgical conditions, blood transfusion, and prognosis of patients receiving and not receiving TXA for coagulopathy.

Variables	All Patients (*n* = 46)	Receiving TXA (*n* = 12)	Not Receiving TXA (*n* = 34)	*p* Value
Median GCS (IQR)	5.5 (4–9)	7.5 (5–10)	5 (4–9)	0.186
DC (%)	23 (50.0)	7 (58.3)	16 (47.1)	0.502
Mean operative blood loss (SD)	649 (499)	733 (481)	619 (509)	0.217
Mean duration of surgery (SD)	167 (98)	162 (68)	168 (108)	0.910
Blood transfusion (%)	36 (78.3)	12 (100.0)	24 (70.6)	0.086
Mean blood transfusion volume (SD)	1053 (1016)	1438 (1235)	917 (909)	0.170
Mean ICU LOS (SD)	5.9 (6.1)	6.5 (4.5)	5.7 (6.7)	0.187
Median GOS (IQR)	3 (1–5)	4 (1–5)	3 (1–4)	0.639
Mortality (%)	17 (37.0)	4 (33.3)	13 (38.2)	0.765

Abbreviations: TXA, tranexamic acid; GCS, Glasgow Coma Scale; DC, decompressive craniectomy; ICU, Intensive Care Unit; LOS, Length of stay; GOS, Glasgow outcome scale; SD, standard deviations; IQR, interquartile range.

**Table 4 jcm-11-00185-t004:** Comparison of functional outcomes at discharge between patients who did and did not develop coagulopathy after open TBI.

	All Patients	Coagulopathy	No Coagulopathy	*p* Value
Favorable functional outcome (GOS score of 4–5)	97 (73.5)	21 (45.7)	76 (88.4)	<0.001
Unfavorable functional outcome (GOS score of 1–3)	35 (26.5)	25 (54.3)	10 (11.6)	

All values are presented as the number of patients (% of total); GOS = Glasgow Outcome Scale.

## Data Availability

The data is available by contacting the corresponding authors.

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
