# Peer review of "Factors Associated with the Development of Coagulopathy after Open Traumatic Brain Injury"

_jcm, 2021, doi:10.3390/jcm11010185_

Round 1

Reviewer 1 Report

The authors reported an interesting study about factors associated with coagulopathy after open TBI.

As they mentioned, identification of risk factors of coagulopathy after TBI is important.

This study was very well-written, but I have several questions.

1. TBI includes epidural, subdural, subarachnoid, or parenchymal hemorrhages. What is the definition of open TBI? I think that the definition of open TBI in this study was not clear. In line 91, dural damage on imaging was defined as open TBI in this study. However, penetrating injury was used in your references (#2, #3), and I don't agree that penetrating injury is equal to dural damage on imaging. Please explain it in detail.

2. The authors said that severe TBI (low GCS score) is a risk factor for coagulopathy. However, patients with coagulopathy tend to have a higher chance of severe TBI after head injury than those without coagulopathy.  I think there may be no causual relationship. What is your opinion?

3. Additional English correction is not required.

Author Response

Response to Reviewer 1 Comments

Dear Editor and Reviewers,

Thanks very much for taking your time to review this manuscript. I really appreciate all your comments and suggestions! Your suggestion has enhanced my manuscript a lot. Please find my itemized responses in below and my revisions/corrections in the re-submitted files.

Point 1: TBI includes epidural, subdural, subarachnoid, or parenchymal hemorrhages. What is the definition of open TBI? I think that the definition of open TBI in this study was not clear. In line 91, dural damage on imaging was defined as open TBI in this study. However, penetrating injury was used in your references (#2, #3), and I don't agree that penetrating injury is equal to dural damage on imaging. Please explain it in detail.

Response 1: In this study, all patients had open TBI. Open TBI is defined as an injury in which the injured object results in the opening of the scalp, skull, dura, and brain tissue to the outside world. If the dura mater is not ruptured and the cranial cavity is not connected to the outside world, the brain injury is still in a closed state. And if there is only skull base fracture, it is called internal opening, otherwise it is external opening. Penetrating injury is a kind of open TBI. Therefore, we excluded patients with evidence of absence of dural damage on imaging. I have added this definition in 89-91 of the revised manuscript.

Point 2: The authors said that severe TBI (low GCS score) is a risk factor for coagulopathy. However, patients with coagulopathy tend to have a higher chance of severe TBI after head injury than those without coagulopathy.  I think there may be no causual relationship. What is your opinion?

Response 2: Thank you for your advice and I am sorry that this part was not stated clearly in the original manuscript. I should explain that we need to describe in the method that patients with a history of coagulopathy, including hypo- and hypercoagulable states, are excluded, and therefore, none of the patients included in this study had pretraumatic coagulopathy. I have corrected it in line 94 of the revised manuscript.

Point 3: Additional English correction is not required.

Response 3: Thank you for your evaluation. This article has been edited by Editage (www.editage.cn) before submission.

Reviewer 2 Report

Authors clearly report that isolated open TBI patients with a low GCS score, high neutrophil/lymphocyte ratio (NLR), low platelet/lymphocyte ratio (PLR), and hyperglycemia at admission were significantly associated with the occurrence of coagulopathy and poor prognosis by investigating multivariate logistic regression analysis in 132 cases.

I have some points to inquire to authors.

  1. The definition of coagulopathy lacks a unified standard as authors described in Limitations. Authors had better add some more references about TBI including other definition of coagulopathy and its outcome.
  2. The definition of Open traumatic injury is unclear. What is Open ?
  3. In Table 1 and 2, some number is out of line. In detail, Table 1, Intracranial fragments, intracranial hematoma and admission glucose. Table 2, intracranial hematoma and admission glucose.

Author Response

Response to Reviewer 2 Comments

Dear Editor and Reviewers,

Thanks very much for taking your time to review this manuscript. I really appreciate all your comments and suggestions! Please find my itemized responses in below and my revisions/corrections in the re-submitted files.

Point 1: The definition of coagulopathy lacks a unified standard as authors described in Limitations. Authors had better add some more references about TBI including other definition of coagulopathy and its outcome.

Response 1: Thank you for your advice and your advice will make my manuscript more colorful. I will add and discuss additional definitions of coagulopathy and the results of its TBI in the revised manuscript, including different cut-offs for different references, resulting in different outcomes. You can see it in lines 358-364 in the revised manuscript.

Point 2: The definition of Open traumatic injury is unclear. What is Open ?

Response 2: In this study, open TBI is defined as an injury in which the injured object results in the opening of the scalp, skull, dura, and brain tissue to the outside world. If the dura mater is not ruptured and the cranial cavity is not connected to the outside world, the brain injury is still in a closed state. And if there is only skull base fracture, it is called internal opening, otherwise it is external opening. I have added this definition in 89-91 of the revised manuscript.

Point 3: In Table 1 and 2, some number is out of line. In detail, Table 1, Intracranial fragments, intracranial hematoma and admission glucose. Table 2, intracranial hematoma and admission glucose.

Response 3: Thank you for your advice, and I'm sorry that this might need to be explained, due to typesetting reasons, the uploaded version leads to some number is out of line, and we have corrected the problem in the revised manuscript.

Reviewer 3 Report

Authors present a retrospective review on 132 patients following severe traumatic brain injury, 46 who developed coagulopathy, in order to determine the clinical characteristics of the coagulopathy following TBI and its relevance for the prognosis. Low GCS, high neutrophile/lymphocyte ratio NLR, low platelet/lymphocyte ratio (PLR) and hyperglycemia at admission were associated with occurence of coagulopathy and poor prognosis; PLR was suggested as a possible novel indicator of coagulopathy in TBI.

The most important issue with this study is that it does not take the important treatment modality of TBI into consideration - the cranial surgery. Also, concomitant injuries (polytraumatic patients, injury of the lung, the spleen) could also play a vital role in development of coagulopathy. Blood loss at surgery should also be included as an independent variable for occurence of coagulation-related problems.

So, as a minimum for further discussion of the results of this study, please include in the revised manuscript: were all patients operated and how were they operated, from how many surgeons and under which diagnosis (there is a substantial difference between surgery for epidural hematoma and decompressive craniectomy due to acute subdural hematoma), what was the mean time from TBI to the surgery, what was the blood loss and duration for surgery (mean values) and did all these parameters differ between patients who developed coagulopathy and those who did not; did they receive blood due to blood loss, how long did the patients spent at the ICU, were they given heparin or other prophylaxis of thrombosis, how many patients were alert and when, what was the follow up.

Please add following to your discussion and comment:

  1. Wada T, Shiraishi A, Gando S, Yamakawa K, Fujishima S, Saitoh D, Kushimoto S, Ogura H, Abe T, Mayumi T, Sasaki J, Kotani J, Takeyama N, Tsuruta R, Takuma K, Shiraishi SI, Shiino Y, Nakada TA, Okamoto K, Sakamoto Y, Hagiwara A, Fujimi S, Umemura Y, Otomo Y. Pathophysiology of Coagulopathy Induced by Traumatic Brain Injury Is Identical to That of Disseminated Intravascular Coagulation With Hyperfibrinolysis. Front Med (Lausanne). 2021 Nov 15;8:767637. doi: 10.3389/fmed.2021.767637. PMID: 34869481; PMCID: PMC8634586.
  2. Bradbury JL, Thomas SG, Sorg NR, Mjaess N, Berquist MR, Brenner TJ, Langford JH, Marsee MK, Moody AN, Bunch CM, Sing SR, Al-Fadhl MD, Salamah Q, Saleh T, Patel NB, Shaikh KA, Smith SM, Langheinrich WS, Fulkerson DH, Sixta S. Viscoelastic Testing and Coagulopathy of Traumatic Brain Injury. J Clin Med. 2021 Oct 28;10(21):5039. doi: 10.3390/jcm10215039. PMID: 34768556; PMCID: PMC8584585. - Please comment on platelet dysfunction as the most important cause for coagulopathy in TBI patients
  3. Wu B, Lu Y, Yu Y, Yue H, Wang J, Chong Y, Cui W. Effect of tranexamic acid on the prognosis of patients with traumatic brain injury undergoing craniotomy: study protocol for a randomised controlled trial. BMJ Open. 2021 Nov 25;11(11):e049839. doi: 10.1136/bmjopen-2021-049839. PMID: 34824110; PMCID: PMC8627390. - please describe your treatment protocoll for patients with coagulopathy in TBI and your thoughts on use of tranexamic acid
  4. Maegele M. Coagulopathy and Progression of Intracranial Hemorrhage in Traumatic Brain Injury: Mechanisms, Impact, and Therapeutic Considerations. Neurosurgery. 2021 Nov 18;89(6):954-966. doi: 10.1093/neuros/nyab358. PMID: 34676410. - please share your thoughts on point of time when to treat coagulopathy and how

Author Response

Response to Reviewer 3 Comments

Dear Editor and Reviewers,

Thanks very much for taking your time to review this manuscript. I really appreciate all your comments and suggestions! Thank you for your suggestions, which add a lot to my article. Please find my itemized responses in below and my revisions/corrections in the re-submitted files.

Point 1: Concomitant injuries (polytraumatic patients, injury of the lung, the spleen) could also play a vital role in development of coagulopathy.  

Response 1: The main subjects of inclusion in this study are patients with isolated open TBI and therefore not accompanied by injuries at other sites, and I have supplemented the instructions for isolated open TBI in line 91 of the revised manuscript.

Point 2: As a minimum for further discussion of the results of this study, please include in the revised manuscript: were all patients operated and how were they operated, from how many surgeons and under which diagnosis (there is a substantial difference between surgery for epidural hematoma and decompressive craniectomy due to acute subdural hematoma),

Response 2: In lines 88-89 of the revised manuscript, you will see that all patients underwent craniotomy, and the operations were performed by the same group of surgeons. According to the different diagnosis, we divide it into decompressive craniectomy and non-decompressive craniectomy in lines 102-103. The coagulopathy group appeared to be more frequently treated with DC (p<0.001).

Point 3: What was the mean time from TBI to the surgery, what was the blood loss and duration for surgery (mean values) and did all these parameters differ between patients who developed coagulopathy and those who did not; did they receive blood due to blood loss, how long did the patients spent at the ICU, were they given heparin or other prophylaxis of thrombosis, how many patients were alert and when, what was the follow up.

Response 3: In lines 101-104 of the revised manuscript, you can see that all these parameters include preoperative tranexamic acid (TXA) treatment, the time from open TBI to the surgery, operative blood loss, duration of surgery, number of blood transfusions , blood transfusion volume, and length of ICU stay (ICU LOS). And statistical analysis is performed in Table 1 and Table 3. The coagulopathy group appeared to be more frequently treated with TXA (p=0.034) and blood transfusion (p<0.001). In addition, patients with coagulopathy required more time for surgery (p=0.029), blood loss (p<0.001), blood transfusion volume (p<0.001), and ICU LOS (p=0.003). The length of time from TBI to surgery did not differ significantly between the two groups (p=0.114). The patient's prognosis and follow-up results can be viewed at 234-237. Patients with coagulopathy have a worse prognosis and higher mortality.

Point 4: Please add following to your discussion and comment: 1. Wada T, Shiraishi A, Gando S, Yamakawa K, Fujishima S, Saitoh D, Kushimoto S, Ogura H, Abe T, Mayumi T, Sasaki J, Kotani J, Takeyama N, Tsuruta R, Takuma K, Shiraishi SI, Shiino Y, Nakada TA, Okamoto K, Sakamoto Y, Hagiwara A, Fujimi S, Umemura Y, Otomo Y. Pathophysiology of Coagulopathy Induced by Traumatic Brain Injury Is Identical to That of Disseminated Intravascular Coagulation With Hyperfibrinolysis. Front Med (Lausanne). 2021 Nov 15;8:767637. doi: 10.3389/fmed.2021.767637. PMID: 34869481; PMCID: PMC8634586.

Response 4: I have discussed  this reference at lines 67-70 of the revised manuscript.

Point 5: 2. Bradbury JL, Thomas SG, Sorg NR, Mjaess N, Berquist MR, Brenner TJ, Langford JH, Marsee MK, Moody AN, Bunch CM, Sing SR, Al-Fadhl MD, Salamah Q, Saleh T, Patel NB, Shaikh KA, Smith SM, Langheinrich WS, Fulkerson DH, Sixta S. Viscoelastic Testing and Coagulopathy of Traumatic Brain Injury. J Clin Med. 2021 Oct 28;10(21):5039. doi: 10.3390/jcm10215039. PMID: 34768556; PMCID: PMC8584585. - Please comment on platelet dysfunction as the most important cause for coagulopathy in TBI patients.

Response 5: I have discussed platelet dysfunction as the most important cause for coagulopathy in TBI patients at lines 303-306 of the revised manuscript. Platelet dysfunction is the most important cause of coagulopathy in TBI patients, including dysfunction of platelet adhesion, activation, and aggregation, but whether platelet transfusion can improve patient outcomes remains controversial.

Point 6: 3. Wu B, Lu Y, Yu Y, Yue H, Wang J, Chong Y, Cui W. Effect of tranexamic acid on the prognosis of patients with traumatic brain injury undergoing craniotomy: study protocol for a randomised controlled trial. BMJ Open. 2021 Nov 25;11(11):e049839. doi: 10.1136/bmjopen-2021-049839. PMID: 34824110; PMCID: PMC8627390. - please describe your treatment protocoll for patients with coagulopathy in TBI and your thoughts on use of tranexamic acid.  4. Maegele M. Coagulopathy and Progression of Intracranial Hemorrhage in Traumatic Brain Injury: Mechanisms, Impact, and Therapeutic Considerations. Neurosurgery. 2021 Nov 18;89(6):954-966. doi: 10.1093/neuros/nyab358. PMID: 34676410. - please share your thoughts on point of time when to treat coagulopathy and how.

Response 6: I have described treatment protocols for patients with coagulopathy in TBI and my thoughts on use of tranexamic acid and shared thoughts of time when to treat coagulopathy and how in lines 325-355 of the revised manuscript. At the same time, statistical analysis is performed in Table 3. I think that early correction of coagulopathy is of great significance for better survival of TBI, and treatment of coagulopathy mainly includes transfusion of fresh frozen plasma (FFP), platelets, and use of TXA. TBI patients with moderate coagulopathy who underwent FFP were more likely to have a poor prognosis. Platelet transfusion has not been shown to improve mortality in patients with intracranial hemorrhage after TBI, but may instead increase transfusion-related risks. However, it is unclear whether patients with significant blood loss and established severe coagulopathy require transfusion, including FFP and platelets. TXA reduces the risk of death related to brain injury in patients with mild to moderate TBI, and early treatment is more effective, but it does not improve the prognosis of patients with severe TBI. Therefore, further studies are needed in the future to confirm the mortality and prognostic impact of blood transfusion and TXA in severe TBI patients with severe coagulopathy.

Round 2

Reviewer 3 Report

The authors have nicely answered the requested questions. I suggest two following minor changes: 1) please add the exact diagnosis of the TBI in procentage (epidural hematoma, acute subdural hematoma); 2) please add one or two illustrative cases of a patient who underwent complications due to coagulopathy with imaging.
